# Prior immunity to *Ureaplasma urealyticum* protects against respiratory infection in immunosuppressed mice

Maha Y. Al-Jabri,[1] Robin Patel,[1,2] Derek Fleming[2]

**ABSTRACT** *Ureaplasma* species can cause systemic infections in immunocompromised hosts, including lung transplant recipients. Here, we investigated the impact of prior exposure to *Ureaplasma urealyticum* on the risk of *U. urealyticum* and *Ureaplasma parvum* infection in mice subjected to an immunosuppression regimen similar to that administered to solid organ transplant recipients. Mice were immunized with three intramuscular injections of *U. urealyticum*, with control mice injected with adjuvant only. Following immunization, mice received tacrolimus, mycophenolate mofetil, and methylprednisolone, and then were challenged with *U. urealyticum* or *U. parvum* intraperitoneally and intratracheally over 6 days. Relative *U. urealyticum* antibody levels in plasma were assessed over time, and lungs were harvested at sacrifice for bacterial load quantification, assessed using a color-changing unit assay. *U. urealyticum* antibody levels were higher in immunized compared with control animals ($P < 0.0001$), even when animals were immunosuppressed. *U. urealyticum* and *U. parvum* lung burden was reduced in immunized compared with control mice (~6 $\log_{10}$ reduction for *U. urealyticum* and <1 $\log_{10}$ reduction for *U. parvum*; $P = 0.008$ and 0.046, respectively). In summary, this study shows that prior exposure to live *U. urealyticum* provides some protection against infection with *U. urealyticum*.

**IMPORTANCE** *Ureaplasma*-induced hyperammonemia syndrome is a rare but potentially deadly complication of solid organ transplantation, especially lung transplantation. The pathophysiology of this relatively recently recognized condition is poorly understood, and it is unclear what factors may influence patient susceptibility. This study investigates the possible protective effects of prior exposure to *Ureaplasma urealyticum* in a mouse model subjected to an immunosuppression regimen similar to that given to lung transplant recipients. The findings show that prior exposure could provide protection against *Ureaplasma* lung infection.

**KEYWORDS** *Ureaplamsa*, lung transplantation, *Ureaplasma*-induced hyperammonemia, immunization

Ureaplasma species can cause invasive infections in immunocompromised patients, including solid organ transplant recipients (1). Numerous case reports note an association between individuals with hypogammaglobulinemia and invasive *Ureaplasma* infections, including septic arthritis, deep abscesses, and endocarditis (2–6). Rarely, *Ureaplasma* species can cause pyelonephritis and pyelitis in kidney transplant recipients (7, 8). In a case series of lung transplant recipients, four of 357 lung transplant recipients developed localized infection with *Ureaplasma urealyticum* alone, resulting in anastomotic dehiscence and ischemia (9). Further, *Ureaplasma* species have been reported to cause hyperammonemia syndrome in lung transplant recipients, characterized by elevated serum ammonia levels with worsening neurologic status (10–13). Although hyperammonemia syndrome only affects 1%–4.1% of lung transplant recipients, it is

**Peer Reviewers** Emily Bryan, The University of Queensland, Brisbane, Australia; Li Xiao, The University of Alabama at Birmingham, Birmingham, Alabama, USA

Address correspondence to Robin Patel, patel.robin@mayo.edu.

R.P. reports grants from MicuRx Pharmaceuticals and BIOFIRE. R.P. is a consultant to PhAST, Day Zero Diagnostics, Abbott Laboratories, Sysmex, DEEPULL DIAGNOSTICS, S.L., Oxford Nanopore Technologies, HealthTrackRx and CARB-X. In addition, R.P. has a patent on *Bordetella pertussis/parapertussis* PCR issued, a patent on a device/method for sonication with royalties paid by Samsung to Mayo Clinic, and a patent on an anti-biofilm substance issued. R.P. receives honoraria from Up-to-Date and the Infectious Diseases Board Review Course. The other authors of this manuscript have no conflicts of interest to disclose as described by the American Journal of Transplantation.

See the funding table on p. 8.

associated with a mortality rate of 57% to 75% if untreated (12, 14, 15). We previously reported that mice immunosuppressed with a pharmacologic regimen similar to that administered to lung transplant recipients, and subsequently challenged with *U. urealyticum* or *U. parvum*, develop *Ureaplasma* infection and associated hyperammonemia (16, 17).

It is incompletely defined why immunocompromised patients, especially lung transplant recipients, are predisposed to *Ureaplasma* infections; understanding predisposing factors can inform prevention strategies. A risk factor in lung transplant recipients is donor allograft *Ureaplasma* colonization; transmission of *Ureaplasma* species from a single donor to more than one organ transplant recipient has been reported (11). Lung transplant recipients whose donor bronchoalveolar fluid tests positive for *Ureaplasma* species are more likely to develop hyperammonemia syndrome than those with negative donor testing (in the absence of preemptive treatment) (18). Another potential risk factor for hyperammonemia is uremia, as suggested in a murine study (19). It is unknown whether immunity to *Ureaplasma* species affects susceptibility to *Ureaplasma* infection. In this study, we investigated the effect of immunity to *U. urealyticum* on development of infection with either *U. urealyticum* or *U. parvum*.

## MATERIALS AND METHODS

### Study isolates

Isolates studied were *U. parvum* (Mayo Clinic clinical isolate IDRL-10744) and *U. urealyticum* (Mayo Clinic clinical isolate IDRL-10611), both from bronchoalveolar lavage fluid of lung transplant recipients with hyperammonemia syndrome; 600 µL aliquots of bacteria suspended in U9 broth (20) buffered with 100 mM 2-ethanesulfonic acid (#475893, EMD Millipore, Billerica, MA) at pH 6 were frozen at −80°C until use. To prepare the inoculum, frozen *Ureaplasma* species were thawed and cultivated in U9 broth in a 5% $CO_2$ incubator at 37°C until a color change from yellow to orange was observed. The suspension was then centrifuged at 12,500 rcf for 30 min to pellet the bacteria, and the pellet resuspended in fresh U9 broth to achieve the concentrations described below.

### Experimental mouse model

Immunocompetent C3H male and female mice (C3H/HeNCrl, Chales River Laboratories, Wilmington, MA) were randomized to immunized and control groups. Immunized mice received a 50 µL intramuscular injection of $10^5$–$10^6$ cells/mL of live *U. urealyticum* suspended in Freund's incomplete adjuvant (#77145, Thermo Scientific, Waltham, MA) (21). Control mice received 50 µL of adjuvant only. Intramuscular injections were administered into the thigh muscles of the hind limb every other week for three doses (Fig. 1).

For *U. urealyticum* challenge, 20 (1 female:1 male) mice were studied (Fig. 1), while for the *U. parvum* challenge, 17 mice were studied, 9 (4 female: 5 male) immunized, and 8 (1 female:1 male) control animals. *U. urealyticum* and *U. parvum* challenges were completed at different times, thus necessitating separate control arms.

Two weeks following the final immunization, mice were pharmacologically immunosuppressed for 13 days, using daily 1.2 mg/kg tacrolimus (Major Pharmaceuticals, Livonia, MI) and 40 mg/kg mycophenolate mofetil (Lannett Company, Trevose, PA) administered via oral gavage, and weekly 0.8 mg methylprednisolone (Amneal Pharmaceuticals, Bridgewater, NJ) administered via subcutaneous injection (Fig. 1). Mice received 40 g/L urea (#U15, Fisher Chemical, Waltham, MA) in their drinking water for the duration of immunosuppression, to mimic uremic conditions (19).

During the last 6 days of immunosuppression, mice were challenged with either *U. urealyticum* or *U. parvum* (Fig. 1), using 100 µL daily intraperitoneal (IP) administration and 50 µL every other day intratracheal (IT) administration of ~$10^6$–$10^7$ CFU/mL of bacteria suspended in saline with 0.1% agar (16, 17). The combination of IP/IT challenges

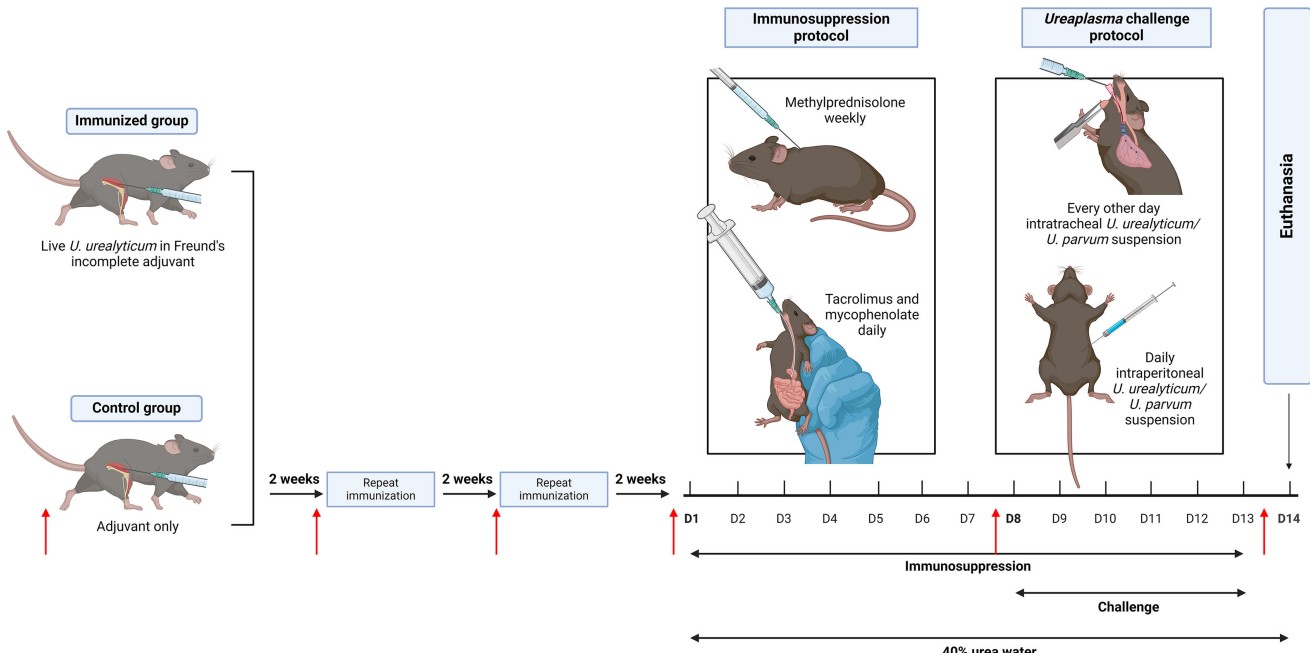

FIG 1 Mouse immunization, immunosuppression, and *Ureaplasma* challenge protocols. Male and female C3H mice were divided into immunized and control groups. Immunized mice received a 50 µL intramuscular injection of $10^5$–$10^6$ cells/mL of live *Ureaplasma urealyticum* suspended in Freund's incomplete adjuvant. Control mice received 50 µL of the adjuvant only. Immunization was carried out every other week for three doses. Two weeks following the last immunization, mice were pharmacologically immunosuppressed for 13 days, using daily tacrolimus and mycophenolate mofetil administered via oral gavage, and weekly methylprednisolone administered via subcutaneous injection. Overlapping the last 6 days of immunosuppression, mice were challenged intraperitoneally and intratracheally with *U. urealyticum* or *Ureaplasma parvum*. Red arrows indicate time points at which blood was drawn. For the *U. urealyticum* challenge, 20 (1 female:1 male) mice were studied, equally distributed between immunized and control animals. For the *U. parvum* challenge, 17 mice were studied, nine (4 female:5 male) immunized, and eight (1 female:1 male) controls. Created using BioRender.com.

was chosen based on past studies showing this methodology to have a high likelihood of generating hyperammonemia in mice (16, 17). Addition of 0.1% agar to the saline solution increases viscosity, mitigating expulsion from the lungs. In prior studies, no ill effects from the 0.1% agar saline vehicle were observed (16, 17, 19, 22). For IT challenge, mice were anesthetized with IP ketamine/xylazine (90/10 mg/kg; Ketaset, Zoetis, Parsippany, NJ, USA/AnaSed LA, VetOne, League City, TX, USA), and the *Ureaplasma* suspension was instilled into the trachea using a 22-gauge curved gavage needle. Mice were placed in a vertical position using a custom-made restraining device for a minimum of 5 min to facilitate drainage of the suspension into the lungs (19).

## *Ureaplasma* antibody (UU-ab) level measurement

Blood was collected via facial vein puncture at multiple intervals throughout the protocol to determine relative UU-ab prior to, during, and following immunization, immunosuppression, and challenge (Fig. 1; red arrows). Relative UU-ab levels were measured in plasma using Abebio Mouse *U. urealyticum* antibody ELISA kits (#AE62674MO, Wuhan Abebio Science Co., Ltd, Wuhan, China), per manufacturer instructions.

## Blood ammonia concentration assessment

Ammonia concentrations, in µmol/L, were measured on whole blood immediately after blood collection via cardiac puncture at the end of the protocol using an Arkray PocketChem BA PA-4140 Blood Ammonia Meter point-of-care kit (ARKRAY America, Inc. Minneapolis, MN, USA). Normal murine blood ammonia concentrations are incompletely defined, but have been reported as 6–71 µmol/L (19, 23).

## Bacterial load quantification

Following harvest, the lungs were weighed (in grams), and homogenized in 1 mL of normal saline. Then, 20 µL of homogenized lung tissue was placed in a 96-well microtiter plate in 180 µL of U9 broth, with subsequent 1:10 dilutions performed in triplicate. The plate was incubated at 37°C and read after 72 h. Change of media color from yellow to fuchsia was assessed, and the well with the lowest dilution where this color change occurred was recorded as the color-changing unit (CCU). $Log_{10}$ CCU/g lung tissue was calculated using the average CCU of the three technical replicates per sample. Blood (cardiac) was also collected for bacterial load quantification via serial dilution and CCU measurement as above.

## Statistical analysis

Data were tested for normality using the Shapiro–Wilk test and for equality of variance using Levene's test (RStudio version 4.2.3, Boston, MA). All other statistical analyses were performed using GraphPad Prism 10 software (Dotmatics, Boston, MA), along with graph generation. Statistical significance of UU-ab levels at different timepoints was calculated using Brown–Forsythe and Welch analysis of variance (ANOVA) tests with Dunnet's T3 multiple comparisons tests, with individual comparisons computed for each pair. Statistical significance of lung bacterial loads and blood ammonia levels between control and immunized mice were calculated using the Mann–Whitney test. Lung infection rates were compared using the $\chi^2$ test.

## RESULTS

### UU-ab in immunized mice compared with control mice

UU-ab levels 2 weeks following the second immunizing dose were higher in immunized compared with control animals (P-value of 0.0003 for *U. urealyticum* challenge and <0.0001 for *U. parvum* challenge, Fig. 2), and remained so 2 weeks following the third immunizing dose (P-value of 0.0005 and 0.0006), as well as following 13 days of immunosuppression and 6 days of infection (P < 0.0001) for both *U. urealyticum* and *U. parvum* challenges (Fig. 2A and B). This demonstrates successful immunization of the study animals.

### *U. urealyticum* load in lungs of *U. urealyticum* immunized mice compared with control mice

For the *U. urealyticum* challenge, the mean lung bacterial burden of *U. urealyticum* was $4.49 \times 10^7$ $log_{10}$ CCU/g in the control group and $3.08 \times 10^1$ $log_{10}$ CCU/g in the immunized group, a ~6 $log_{10}$ reduction (P = 0.008, Fig. 3). The overall infection rate (i.e., detection of *U. urealyticum* in the lungs) of control mice was 80% compared with 30% in immunized mice (P = 0.0246). This demonstrates that *U. urealyticum* immunization protects, at least partially, against *U. urealyticum* infection. All blood cultures were negative, suggesting a lack of systemic infection in this model.

### *U. parvum* load in the lungs of *U. urealyticum* immunized mice compared with control mice

In the *U. parvum*-challenged animals, the mean lung bacterial burden of *U. parvum* was $1.19 \times 10^8$ $log_{10}$ CCU/g in the control group and $3.79 \times 10^7$ $log_{10}$ CCU/g *U. urealyticum* in the immunized group, a less than 1 $log_{10}$ reduction (P = 0.046, Fig. 3). The lung infection rate was 100% in both immunized and control groups. This demonstrates that *U. urealyticum* immunization provides much less protection against *U. parvum* than against *U. urealyticum* infection.

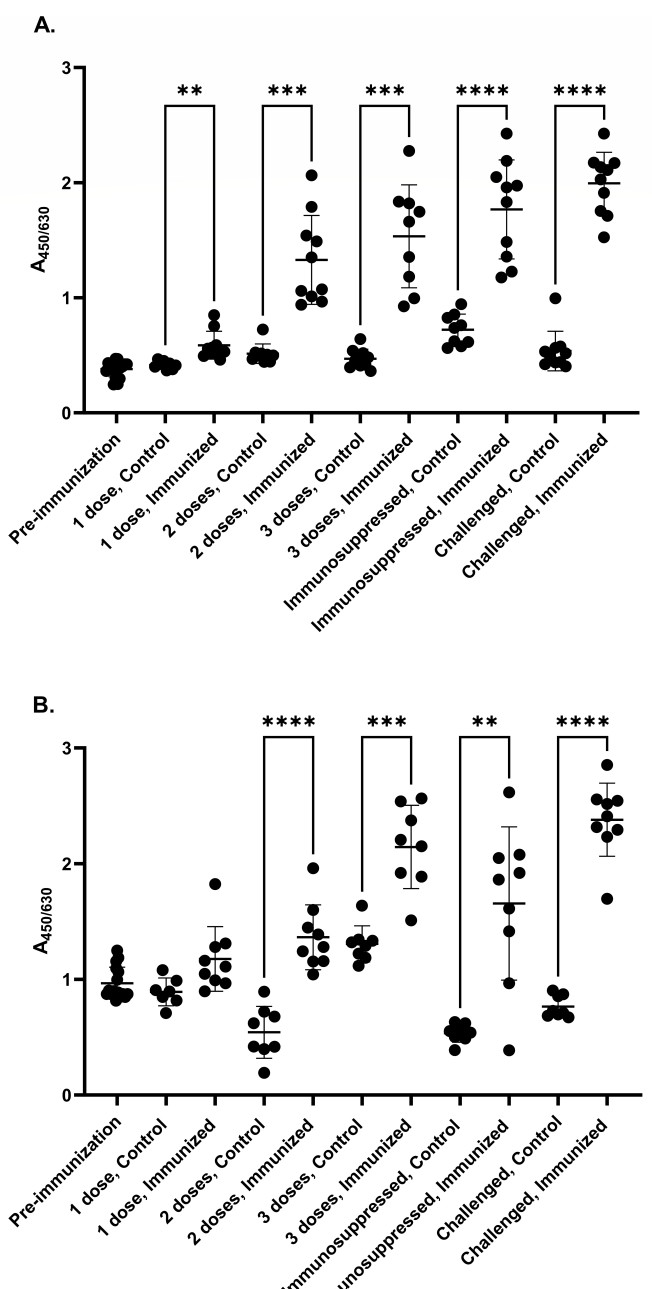

**FIG 2** Relative plasma UU-ab levels (ELISA; 450 nm/630 nm) in the *Ureaplasma urealyticum* (A) and *Ureaplasma parvum* (B) challenges. Timepoints are pre-immunization, 2 weeks following 1, 2, and 3 immunization doses, following 7 days of immunosuppression, and following 13 days of immunosuppression and 6 days of *U. urealyticum* and *U. parvum* challenge. Controls were immunized with adjuvant only. $N = 10$ (*U. urealyticum* control); $N = 10$ (*U. urealyticum* immunized); $N = 8$ (*U. parvum* control); $N = 9$ (*U. parvum* immunized). Graphs show individual data points with means represented in bold horizontal lines, and error bars representing standard deviations. Brown–Forsythe and Welch ANOVA tests with Dunnet's T3 multiple comparisons tests were used, with individual comparisons computed for each pair; $**P \leq 0.01$; $***P \leq 0.001$; $****P \leq 0.0001$. GraphPad Prism 10 was used to generate statistics and to plot the graph.

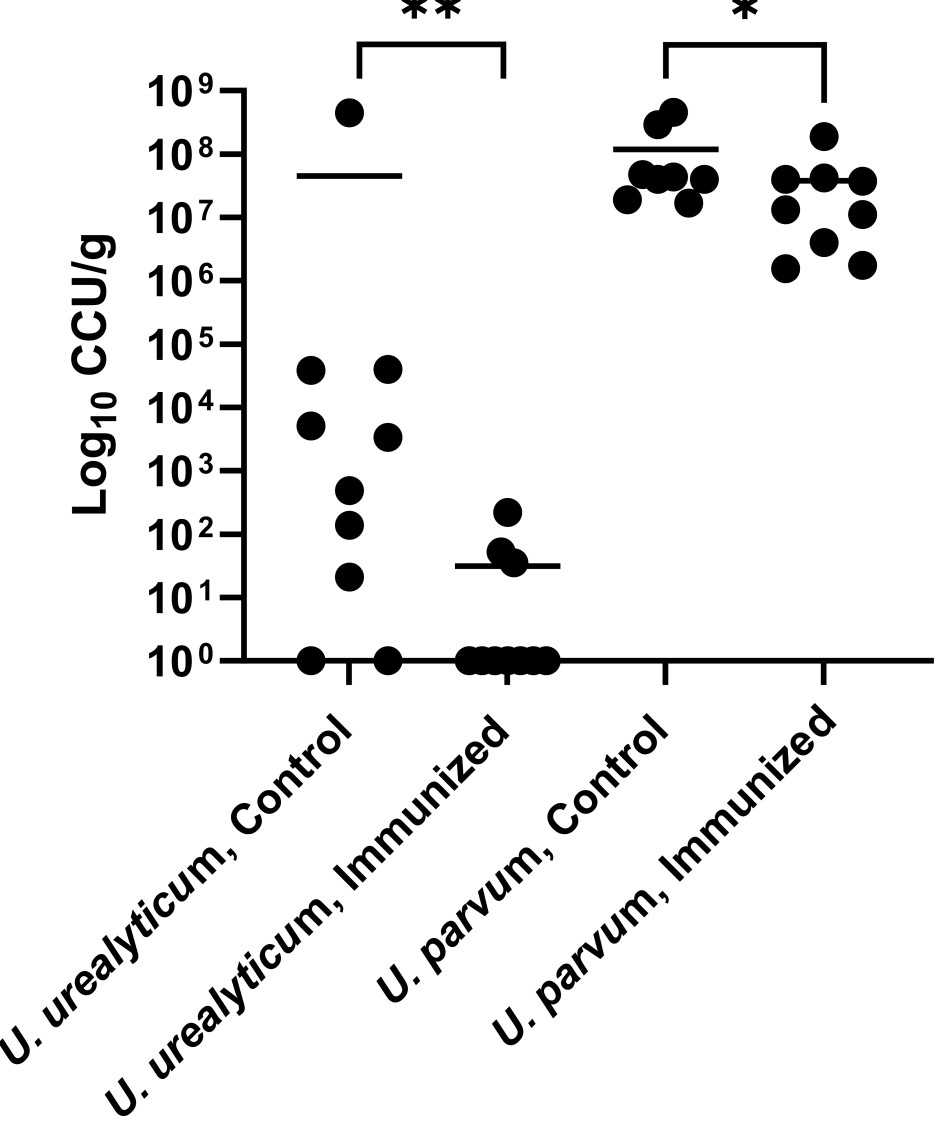

**FIG 3** *Ureaplasma urealyticum* and *Ureaplasma parvum* bacterial load in the lungs of immunized and control mice, measured in $\log_{10}$ color-changing units per gram ($\log_{10}$ CCU/g) of lung tissue. The lungs were harvested after 6 days of the start of challenge protocol. *N* = 10 (*U. urealyticum* control); *N* = 10 (*U. urealyticum* immunized); *N* = 8 (*U. parvum* control); *N* = 9 (*U. parvum* immunized). Graphs show individual data points with means represented in bold horizontal lines. Mann–Whitney; *\*P* = 0.046; *\*\*P* = 0.008. GraphPad prism 10 was used to generate statistics and to plot the graph.

## Blood ammonia levels in immunized and control mice infected with *U. urealyticum* or *U. parvum*

In *U. urealyticum*-challenged animals, the mean blood ammonia concentration was 65 µmol/L (standard deviation [SD] 83) in control and 50 µmol/L (SD 41) immunized mice (Fig. S1). In *U. parvum*-challenged animals, the mean blood ammonia concentration was 12 µmol/L (SD 10) in control and 25 µmol/L (SD 36) in immunized mice (Fig. S1). A single female mouse in the control group that received *U. urealyticum* challenge required euthanasia 3 days prior to the end of the protocol due to illness/stress and cachexia; this animal had a mass anterior to the rib cage, of unknown significance. Analysis of *U. urealyticum* lung bacterial burden in this animal showed >$10^8$ $\log_{10}$ CCU/g and a blood ammonia concentration above the upper limit of quantification (>284 µmol/L), suggesting that this single control animal had *Ureaplasma*-associated hyperammonemia.

It was not possible to assess the effect of immunization on *Ureaplasma*-associated hyperammonemia, given that only a single control animal had documented hyperammonemia.

## DISCUSSION

*Ureaplasma* species are rare causes of disease, whether localized or disseminated, in immunocompromised hosts (9, 14, 15, 24). Risk factors for development of *Ureaplasma*-related disorders, beyond donor colonization in lung transplant recipients, are poorly defined.

Here, the effect of *Ureaplasma* immunity generated by *U. urealyticum* immunization reinforced with Freund's incomplete adjuvant on the risk of *Ureaplasma* infection was evaluated. Immunization resulted in increased plasma UU-ab levels. Although three doses were given, two may have been sufficient to elicit immunogenicity, as immunized mice had 2.5-fold higher UU-ab titers compared with controls 2 weeks following the second immunizing dose. Further, there was no effect of immunosuppression on titers (Fig. 2A and B). This indicates a strong, sustained antibody response.

Additionally, the study investigated the ability of prior exposure to *U. urealyticum* in protecting against future infections. In *U. urealyticum* immunized mice subsequently challenged with the same strain of *U. urealyticum*, lung bacterial loads were markedly lower than control mice (~6 $\log_{10}$ CCU/g reduction, Fig. 3). Lung infection rates for *U. urealyticum* were 80% for control and 30% for immunized mice. The lack of universal infection in all challenged controls could be due to failure of the protocol to achieve infection in all cases, or spontaneous clearance of infection. Alternative strategies, such as use of other adjuvants, or different immunization sites and timing, could be explored to increase protection rates in immunized mice.

This study also addressed whether the level of protection against *Ureaplasma* species is species- (or strain-) specific. When challenged with *U. parvum,* although lung bacterial loads were statistically lower in immunized compared with control mice, the difference was small (<1 $\log_{10}$ CCU/g reduction, Fig. 3), with *U. parvum* challenge infection rates of 100% for both control and immunized mice. This suggests that protective immunity may be species- (or possibly strain-) specific, with minimal, if any, cross-protection between *Ureaplasma* species. Notably, *U. parvum* lung bacterial loads were higher than *U. urealyticum* bacterial loads in control mice, despite the same inoculum being used. Further studies are needed to examine immune factors that could account for a decrease in cross-species (or cross-strain) protection.

*Ureaplasma* species are part of the normal genitourinary microbiota in a proportion of the healthy population (25–30). Conceivably, natural exposure to these agents over time may generate protective immunity in individuals subsequently challenged with *Ureaplasma* species while immunosuppressed. Assessment of the impact of prior immunity to *Ureaplasma* species on risk of *Ureaplasma* infection in immunocompromised patients is needed. If an association was to be confirmed in humans, assessment of immunity could be used to identify those needing close monitoring for *Ureaplasma* infection or possibly prophylaxis, as is done with other infectious agents, such as cytomegalovirus. Development of a human vaccine could also be considered, but *Ureaplasma* infections are rare (31). Cost *versus* benefit of a vaccine for certain transplant populations (e.g., all lung transplant candidates) or targeted to "at-risk" transplant populations (e.g., lung transplant candidates with no evidence of *Ureaplasma* immunity) would need to be considered.

There are several limitations of this study. Although we previously used a version of the described animal model to induce *Ureaplasma*-associated hyperammonemia, only one animal developed this complication in the current study, and that animal was prematurely euthanized. This precluded specific assessment of the effect of *Ureaplasma* immunization on *Ureaplasma*-associated hyperammonemia (Fig. S1). In prior studies from our group, mice developed hyperammonemia with a similar infection protocol (16, 17). Due to the long immunization protocol used here, however, animals were at

least 2 months older than in prior studies, with higher average body weights, which may have impacted susceptibility to *Ureaplasma*-associated hyperammonemia. Another limitation is that naturally acquired immunity may differ from the adjuvant-enhanced immunity generated from immunizations administered in this study. Also, only antibody-mediated immunity was explored here. Exposure to *U. urealyticum* may trigger innate or cell-mediated immunity not accounted for here. Given the limited current understanding of host immunity against *Ureaplasma* species, mechanisms of immune protection against *Ureaplasma* infection deserve further study (32). Further, only relative plasma antibody levels comparisons were generated. Additionally, the antibody assay used does not differentiate between antibody classes; it is not possible to know if only IgM was present throughout the study period, or if class switching took place. Finally, due to its prevalence in cases of hyperammonemia syndrome (9, 11, 13), only *U. urealyticum* prior exposure was studied, representing another limitation. Prior exposure with *U. parvum* may offer different levels of protection, including cross-species protection.

In conclusion, the results of this study show that, in a mouse model, immunization with live *U. urealyticum* provides at least partial protective immunity against *U. urealyticum*, and minimal cross-species protection against *U. parvum* infection.

## ACKNOWLEDGMENTS

The research reported herein was supported by the National Institute of Allergy and Infectious Diseases of the National Institutes of Health under award number R21AI150649. The content is solely the responsibility of the authors and does not necessarily represent the official views of the National Institutes of Health.

## AUTHOR AFFILIATIONS

[1]Divisin of Public Health, Infectious Diseases, and Occupational Medicine, Department of Medicine, Mayo Clinic, Rochester, Minnesota, USA
[2]Division of Clinical Microbiology, Department of Laboratory Medicine and Pathology, Mayo Clinic, Rochester, Minnesota, USA

## AUTHOR ORCIDs

Maha Y. Al-Jabri ⓘ http://orcid.org/0009-0009-4990-0476
Robin Patel ⓘ http://orcid.org/0000-0001-6344-4141
Derek Fleming ⓘ https://orcid.org/0000-0002-0054-904X

## FUNDING

| Funder | Grant(s) | Author(s) |
| --- | --- | --- |
| HHS \| NIH \| National Institute of Allergy and Infectious Diseases (NIAID) | R21AI150649 | Robin Patel |

## AUTHOR CONTRIBUTIONS

Maha Y. Al-Jabri, Conceptualization, Data curation, Formal analysis, Investigation, Methodology, Writing – original draft, Writing – review and editing | Robin Patel, Conceptualization, Funding acquisition, Methodology, Supervision, Writing – review and editing | Derek Fleming, Conceptualization, Data curation, Formal analysis, Investigation, Methodology, Supervision, Writing – review and editing

## DATA AVAILABILITY STATEMENT

Data from this study will be made available to investigators after approval of a data use proposal. Requests may be submitted to patel.robin@mayo.edu.

## ETHICS APPROVAL

This study was conducted following the recommendations outlined in the Guide for the Care and Use of Laboratory Animals by the National Institutes of Health and received approval from the Mayo Clinic Institutional Animal Care and Use Committee (protocol number: A5004-20-R22). Mayo Clinic maintains AAALAC accreditation (000717), USDA registration (41R-0006), and an Assurance with OLAW (A3291-01). Mice were housed in a biosafety level 2, specific-pathogen-free, AAALAC-accredited facility, with sentinel mice tested quarterly for murine pathogens; all tests confirmed the absence of murine pathogens throughout the study. The animals had unrestricted access to irradiated rodent food (LabDiet formula 5053) and water. Environmental conditions in the housing room were controlled (temperature 68°F–74°F, relative humidity 30%–70%, 12:12 hour light–dark cycle). Efforts were made to minimize animal suffering, with mice monitored daily and anesthetized mice observed until fully awake. Monitoring included assessment for decreased activity, lowered body temperature, hunched posture, signs of distress, and difficulty eating or drinking; one mouse required premature euthanasia.

## ADDITIONAL FILES

The following material is available online.

### Supplemental Material

**Figure S1 (Spectrum01763-24-s0001.pdf).** Blood ammonia concentrations in *Ureaplasma urealyticum*- and *Ureaplasma parvum*-infected control and immunized mice.

### Open Peer Review

**PEER REVIEW HISTORY (review-history.pdf).** An accounting of the reviewer comments and feedback.

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
