## [Reviewer comments · Microbiology Spectrum]

Microbiology Spectrum

Prior immunity to *Ureaplasma urealyticum* protects against respiratory infection in immunosuppressed mice

Maha Al-Jabri, Robin Patel, and Derek Fleming

Corresponding Author(s): Robin Patel, Mayo Clinic Minnesota

Review Timeline:

Submission Date:	July 23, 2024
Editorial Decision:	September 4, 2024
Revision Received:	October 28, 2024
Accepted:	October 28, 2024

Editor: Artem Rogovsky

Reviewer(s): Disclosure of reviewer identity is with reference to reviewer comments included in decision letter(s). The following individuals involved in review of your submission have agreed to reveal their identity: Emily Bryan (Reviewer #1); Li Xiao (Reviewer #2)

Transaction Report:

DOI: <https://doi.org/10.1128/spectrum.01763-24>

Re: Spectrum01763-24 (Prior immunity to *Ureaplasma urealyticum* prevents *Ureaplasma urealyticum* infection and, to a lesser extent, *Ureaplasma parvum* infection, in immunosuppressed mice)

Dear Dr. Robin Patel:

Thank you for the privilege of reviewing your work. Below you will find my comments, instructions from the Spectrum editorial office, and the reviewer comments.

Revision Guidelines

Sincerely,
Artem Rogovsky
Editor
Microbiology Spectrum

Reviewer #1 (Comments for the Author):

Summary: The authors examine the antibody response to immunisation with whole, live *Ureaplasma urealyticum*. Next, authors challenged with both UU and UP and measure clearance from lung tissue, in immune suppressed conditions, modelling the setting of a transplant recipient. The authors have provided evidence that vaccine-mediated protection against *Ureaplasma* infection may be possible, and either this or determination of prior immunity could be used as a measure to protect transplant

recipients from the rare but often fatal consequences of transplant-borne infection.

Abstract: Well written and descriptive. The 'importance' section raises several questions, it might be worth clarifying or commenting on the use of the live pre-challenge, compared to e.g. live-attenuated or UV-inactivated, which are common vaccine components. A competent/intact live infection is unlikely to make a viable vaccine component.

Introduction: Well written and covers useful topics. Can the authors comment on what is currently known about immunity to *Ureaplasma*, e.g. which types of innate/cellular/humoral responses are required? Can the authors also comment on the current state of immunisation trials for *Ureaplasma*. This is needed to contextualise the current study in the existing literature.

Materials and Methods:

Line 73: Please list the origin of the isolates used, can just be a reference to previous work for example.

Line 75: Please provide a reference or full recipe details for U9 media.

Line 76 and anywhere else: Please provide the product details for all products listed in the section (catalogue number, brand, city/state, country).

Line 78 and anywhere else: Please provide centrifugation speeds in RCF (xg), not RPM, so methods are reproducible.

Line 83: Please indicate the site of IM injection, e.g. hind-limb?, or provide a reference.

Line 83: Why was such a large range (10-fold) of UU given, could this not be quantified to a smaller range?

What was the reasoning for the intraperitoneal infection? What natural infection/transplant infection does this represent? Can a comment be added to contextualise this? Does this cause systemic infection, which may impact the host response to the IT infection?

What was the purpose of including 0.1% agar in the IT infection? Can you provide a reference for this method demonstrating that it does not cause respiratory distress or inflammation that impacts infection? As there is no vehicle control for infection, this is important.

Line 98: Was Ket/xyl administered IP? And please provide product details.

As the ethics section is quite short, can the authors please add a short statement on whether the mice were monitored for adverse outcomes after all challenges?

Line 100: Does the custom made restraint have a reference? If not, perhaps a supplementary figure with a diagram or photo would be appropriate.

Line 102: how was blood collected? E.g. saphenous or tail-vein bleed? And at which intervals?

Please clarify that lungs were weighed before homogenisation, to obtain the /g part of the measurement.

Was data tested to determine the best statistical test, e.g. normality, outlier, mean and standard deviation comparison, etc. If so, please state this. If not, please conduct these tests to ensure correct statistical tests were applied.

Results:

Line 138: Although there is an impressive reduction in infection rate, I would suggest that with still 30% of animals infected, this is indicative of partial protection.

I would also suggest editing the whole manuscript to use the same terminology throughout, when describing immunisation, challenge, and infection, to ensure terms are not used interchangeably.

Line 145: It is arguable that having a 100% infection rate in both groups, after having demonstrated that UU infection can be cleared, that this result indicates immunisation is actually not protective for UP, contrary to the authors claims. The <1 log decrease in burden indicates partial protection, at best. Suggesting rewriting this section to temper the authors claims.

Similar comment for the title of the manuscript. In addition, for the manuscript title, I am suggesting that this overreaches based on the data. 30% of mice were not immune, and when we consider that up to 20% mice may have failed to be infected/spontaneously cleared as seen in the control group, this value could be higher, as these mice would have been considered immune based on the study parameters. Suggest rewriting the title to more accurately reflect study outcomes.

Discussion:

As above, the language around vaccine mediated protection should be tempered to accurately reflect the data.

The second paragraph of the discussion, starting line 172, seems to just be a restating of results and can be removed or shortened to make space for other discussion points, potentially around the final sentence on further study.

Line 186: It would be worth mentioning that in natural infection, *Ureaplasma* have immune evasion mechanisms, e.g. MBA variation. The difference(s) between naturally-acquired and vaccine-mediated immunity may be vast.

Can the authors comment on the relevance and impact of the infectious challenge used, compared to clinically identified infectious burden?

Can the authors comment on what strategies they might take to achieve protection in 100% of infections, e.g. different adjuvant use, specific antigen use, etc. Or alternatively, provide some discussion about why 30% of animals were not protected?

Figure 3: I'm not convinced that 'sham vaccinated' is the correct term here, as opposed to 'adjuvant only', which I think was the term used earlier in the text. As there was no inclusion of e.g. *Ureaplasma* growth media, to measure the effects of a sham immunisation, e.g. immunogenicity of the media.

Additionally, can the authors provide evidence, e.g. statistical analysis, that control and immunised antibody measurements are significantly increased above pre-vaccination levels.

Please do a manuscript-wide proof-read to make sure abbreviations for UU and UP are consistent. E.g. in Figure 3, the

difference between *U. urealyticum* and *Ureaplasma parvum* and Figure 4 where both are in full.

Figure 4: can the authors please add a small amount more detail to the legend, e.g. what timepoint post infection tissue was harvested.

For graph-based figures, please include the program (version) that graphs and statistics were produced in.

Somewhere in the methods or figure legends, the authors need to indicate what graphs represent in terms of mean vs median, and SD vs SEM. It would also be beneficial to include error bars on graphs. It is unclear why a bar chart has been used in Figure 3, but not 4. However, this is a good use of displaying individual data points. Potentially a box and whisker plot would solve this issue, as it can be used to display (i) individual data points overlaid on summary data (the box), (ii) group mean, and (iii) error bars.

Reviewer #2 (Comments for the Author):

The authors reported study on the protection of *Ureaplasma* species infection after immunization with *U. urealyticum* in a mouse model and found a better protection with the same species (strain) *U. urealyticum* immunization. This is an interesting study that provided possible directions in control the *Ureaplasma*-induced hyperammonemia syndrome in solid organ transplantation. The major concern about this report is the study design. Only *U. urealyticum* was used for immunization while *U. parvum*, which is a more common colonizer in humans was left, making the whole study looks like to be accomplished by half. The experimental grouping strategy looked confusing. It would also be interesting to test more strains within each species for infection to verify the specificity range of the immunity.

Other comments:

1. Line 78: "12,500 rpm". Please provide the corresponding number of g forces.
2. The description of the mouse vaccination and infection protocol is not clear and confusing. Were *U. urealyticum* and *U. parvum* infection protocols carried out the same time? If yes, why did they have different controls? It seems all 37 animals could be better organized for grouping. Figure 1 and Figure 2 may be combined to show the whole experimental procedures. The time points when blood was drawn can be labeled in the combined figure to help the readers better understand Figure 3. In Figure 1, what's the meaning of "10x"? "1:1" can be deleted and instead, the exact numbers of male and female animals should be shown in each group.
3. For statistical analysis, please confirm if a normality test was performed prior to each designated statistical test.
4. For all differences in *U. urealyticum* antibody levels, bacterial loads, blood ammonia levels, are there any differences between male and female?
5. In Figure 3, the *U. urealyticum* antibody levels before infection in the two protocols are different. It looks like the background level (pre-vaccination) in panel B is about 2x of that in panel A. Are there any explanations for this phenomenon? Please also indicate the statistical significance test result between other group combinations, such as different doses groups, and immunosuppressed vs different doses.
6. What's the possible reason for the fact that bacterial loads of *U. parvum* were higher than *U. urealyticum* in control mice?

Summary: The authors examine the antibody response to immunisation with whole, live *Ureaplasma urealyticum*. Next, authors challenged with both UU and UP and measure clearance from lung tissue, in immune suppressed conditions, modelling the setting of a transplant recipient. The authors have provided evidence that vaccine-mediated protection against *Ureaplasma* infection may be possible, and either this or determination of prior immunity could be used as a measure to protect transplant recipients from the rare but often fatal consequences of transplant-borne infection.

Abstract: Well written and descriptive. The 'importance' section raises several questions, it might be worth clarifying or commenting on the use of the live pre-challenge, compared to e.g. live-attenuated or UV-inactivated, which are common vaccine components. A competent/intact live infection is unlikely to make a viable vaccine component.

Introduction: Well written and covers useful topics. Can the authors comment on what is currently known about immunity to *Ureaplasma*, e.g. which types of innate/cellular/humoral responses are required? Can the authors also comment on the current state of immunisation trials for *Ureaplasma*. This is needed to contextualise the current study in the existing literature.

Materials and Methods:

Line 73: Please list the origin of the isolates used, can just be a reference to previous work for example.

Line 75: Please provide a reference or full recipe details for U9 media.

Line 76 and anywhere else: Please provide the product details for all products listed in the section (catalogue number, brand, city/state, country).

Line 78 and anywhere else: Please provide centrifugation speeds in RCF (xg), not RPM, so methods are reproducible.

Line 83: Please indicate the site of IM injection, e.g. hind-limb?, or provide a reference.

Line 83: Why was such a large range (10-fold) of UU given, could this not be quantified to a smaller range?

What was the reasoning for the intraperitoneal infection? What natural infection/transplant infection does this represent? Can a comment be added to contextualise this? Does this cause systemic infection, which may impact the host response to the IT infection?

What was the purpose of including 0.1% agar in the IT infection? Can you provide a reference for this method demonstrating that it does not cause respiratory distress or inflammation that impacts infection? As there is no vehicle control for infection, this is important.

Line 98: Was Ket/xyl administered IP? And please provide product details.

As the ethics section is quite short, can the authors please add a short statement on whether the mice were monitored for adverse outcomes after all challenges?

Line 100: Does the custom made restraint have a reference? If not, perhaps a supplementary figure with a diagram or photo would be appropriate.

Line 102: how was blood collected? E.g. saphenous or tail-vein bleed? And at which intervals?

Please clarify that lungs were weighed before homogenisation, to obtain the /g part of the measurement.

Was data tested to determine the best statistical test, e.g. normality, outlier, mean and standard deviation comparison, etc. If so, please state this. If not, please conduct these tests to ensure correct statistical tests were applied.

Results:

Line 138: Although there is an impressive reduction in infection rate, I would suggest that with still 30% of animals infected, this is indicative of partial protection.

I would also suggest editing the whole manuscript to use the same terminology throughout, when describing immunisation, challenge, and infection, to ensure terms are not used interchangeably.

Line 145: It is arguable that having a 100% infection rate in both groups, after having demonstrated that UU infection can be cleared, that this result indicates immunisation is actually not protective for UP, contrary to the authors claims. The <1 log decrease in burden indicates partial protection, at best. Suggesting rewriting this section to temper the authors claims.

Similar comment for the title of the manuscript. In addition, for the manuscript title, I am suggesting that this overreaches based on the data. 30% of mice were not immune, and when we consider that up to 20% mice may have failed to be infected/spontaneously cleared as seen in the control group, this value could be higher, as these mice would have been considered immune based on the study parameters. Suggest rewriting the title to more accurately reflect study outcomes.

Discussion:

As above, the language around vaccine mediated protection should be tempered to accurately reflect the data.

The second paragraph of the discussion, starting line 172, seems to just be a restating of results and can be removed or shortened to make space for other discussion points, potentially around the final sentence on further study.

Line 186: It would be worth mentioning that in natural infection, *Ureaplasma* have immune evasion mechanisms, e.g. MBA variation. The difference(s) between naturally-acquired and vaccine-mediated immunity may be vast.

Can the authors comment on the relevance and impact of the infectious challenge used, compared to clinically identified infectious burden?

Can the authors comment on what strategies they might take to achieve protection in 100% of infections, e.g. different adjuvant use, specific antigen use, etc. Or alternatively, provide some discussion about why 30% of animals were not protected?

Figure 3: I'm not convinced that 'sham vaccinated' is the correct term here, as opposed to 'adjuvant only', which I think was the term used earlier in the text. As there was no inclusion of e.g. Ureaplasma growth media, to measure the effects of a sham immunisation, e.g. immunogenicity of the media.

Additionally, can the authors provide evidence, e.g. statistical analysis, that control and immunised antibody measurements are significantly increased above pre-vaccination levels.

Please do a manuscript-wide proof-read to make sure abbreviations for UU and UP are consistent. E.g. in Figure 3, the difference between U. urealyticum and Ureaplasma parvum and Figure 4 where both are in full.

Figure 4: can the authors please add a small amount more detail to the legend, e.g. what timepoint post infection tissue was harvested.

For graph-based figures, please include the program (version) that graphs and statistics were produced in.

Somewhere in the methods or figure legends, the authors need to indicate what graphs represent in terms of mean vs median, and SD vs SEM. It would also be beneficial to include error bars on graphs. It is unclear why a bar chart has been used in Figure 3, but not 4. However, this is a good use of displaying individual data points. Potentially a box and whisker plot would solve this issue, as it can be used to display (i) individual data points overlaid on summary data (the box), (ii) group mean, and (iii) error bars.

Reviewer #1 (Comments for the Author):

Abstract: Well written and descriptive. The 'importance' section raises several questions, it might be worth clarifying or commenting on the use of the live pre-challenge, compared to e.g. live-attenuated or UV-inactivated, which are common vaccine components. A competent/intact live infection is unlikely to make a viable vaccine component.

- **Our study aim was to create a “prior infection” model, not a vaccine model; this has been clarified throughout.**

Introduction: Well written and covers useful topics. Can the authors comment on what is currently known about immunity to *Ureaplasma*, e.g. which types of innate/cellular/humoral responses are required? Can the authors also comment on the current state of immunisation trials for *Ureaplasma*. This is needed to contextualise the current study in the existing literature.

Current understanding of immunity to *Ureaplasma* is limited; this has now been addressed in the Discussion. No current trials for *Ureaplasma* vaccines are available to the best of the authors' knowledge.

Materials and Methods:

Line 73: Please list the origin of the isolates used, can just be a reference to previous work for example.

- **Done (Lines 71-73).**

Line 75: Please provide a reference or full recipe details for U9 media.

- **Done.**

Line 76 and anywhere else: Please provide the product details for all products listed in the section (catalogue number, brand, city/state, country).

- **Added.**

Line 78 and anywhere else: Please provide centrifugation speeds in RCF (xg), not RPM, so methods are reproducible.

- **This was an error. 12,500 was the RCF, not the RPM. This has been corrected.**

Line 83: Please indicate the site of IM injection, e.g. hind-limb?, or provide a reference.

- **Done.**

Line 83: Why was such a large range (10-fold) of UU given, could this not be quantified to a smaller range?

- **Given the inherent challenges in *Ureaplasma* culture (lack of turbidity due to small cell size, long culture times, alkalization of the media leading to rapid bacterial loss, etc.) we were unable to narrow the inoculation dose range further. However, all immunized mice received the same bacterial preparation at each immunization time-point.**

What was the reasoning for the intraperitoneal infection? What natural infection/transplant infection does this represent? Can a comment be added to contextualise this? Does this cause systemic infection, which may impact the host response to the IT infection?

- **The protocol is based on prior study (doi: 10.1371/journal.pone.0161214.), in which different infection modalities were compared, and a combination approach of intraperitoneal plus intratracheal infection resulted in mice with the highest serum ammonia levels. We relied on those results when selecting the infection modality that has the highest likelihood of inducing hyperammonemia. Blood cultures were negative, suggesting no systemic infection. Clarification has been added to the text (Lines 100-101).**

What was the purpose of including 0.1% agar in the IT infection? Can you provide a reference for this method demonstrating that it does not cause respiratory distress or inflammation that impacts infection? As there is no vehicle control for infection, this is important.

- **Addition of 0.1% agar to the saline vehicle increases viscosity, decreasing expulsion from the lungs. In past studies, vehicle control mice had no ill effect from the 0.1% agar solution. Clarification has been added to the text (Lines 102-103).**

Line 98: Was Ket/xyl administered IP? And please provide product details.

- **Yes, they were administered IP. Product details have been added.**

As the ethics section is quite short, can the authors please add a short statement on whether the mice were monitored for adverse outcomes after all challenges?

- **The ethics statement has been made more comprehensive.**

Line 100: Does the custom made restraint have a reference? If not, perhaps a supplementary figure with a diagram or photo would be appropriate.

- **The reference has been added.**

Line 102: how was blood collected? E.g. saphenous or tail-vein bleed? And at which intervals?

- **Blood collected via the facial vein. Intervals have been added to Figure 1 (red arrows).**

Please clarify that lungs were weighed before homogenisation, to obtain the /g part of the measurement.

- **Clarified in the text.**

Was data tested to determine the best statistical test, e.g. normality, outlier, mean and standard deviation comparison, etc. If so, please state this. If not, please conduct these tests to ensure correct statistical tests were applied.

- **Based on suggestions from both reviewers, all data was analyzed for normal distribution via the Shapiro-Wilk test, and for equal variance via**

Levene's test. For Figure 2 (relative plasma antibody levels), while the data was normally distributed, the variances were not equal. Therefore, data was re-analyzed using Brown-Forsythe and Welch ANOVA tests with Dunnett's T3 multiple comparisons test, with individual comparisons computed for each comparison. The figure, figure legend, and manuscript text have been updated accordingly. For Supplementary Figure, the data was not normally distributed, and thus the Mann-Whitney test was used in place of the unpaired t-test. While the figure is unchanged, p-values have been updated. Notably, the updated statistical tests resulted in no changes in data interpretation throughout the manuscript.

Results:

Line 138: Although there is an impressive reduction in infection rate, I would suggest that with still 30% of animals infected, this is indicative of partial protection.

- **Statement changed to “This demonstrates that *U. urealyticum* immunization protects, at least partially, against *U. urealyticum* infection”.**

I would also suggest editing the whole manuscript to use the same terminology throughout, when describing immunisation, challenge, and infection, to ensure terms are not used interchangeably.

- **Manuscript and figures have been edited to unify terminology.**

Line 145: It is arguable that having a 100% infection rate in both groups, after having demonstrated that UU infection can be cleared, that this result indicates immunisation is actually not protective for UP, contrary to the authors claims. The <1 log decrease in burden indicates partial protection, at best. Suggesting rewriting this section to temper the authors claims.

- **We agree with the reviewer. The last sentence of section 3.3 was modified to “This demonstrates that *U. urealyticum* immunization provides little to no protection against *U. parvum* infection.”**

Similar comment for the title of the manuscript. In addition, for the manuscript title, I am suggesting that this overreaches based on the data. 30% of mice were not immune, and when we consider that up to 20% mice may have failed to be infected/spontaneously cleared as seen in the control group, this value could be higher, as these mice would have been considered immune based on the study parameters. Suggest rewriting the title to more accurately reflect study outcomes.

- **The title has been revised to “Prior immunity to *Ureaplasma urealyticum* protects against respiratory infection in immunosuppressed mice”.**

Discussion:

As above, the language around vaccine mediated protection should be tempered to accurately reflect the data.

- **The language has been altered accordingly.**

The second paragraph of the discussion, starting line 172, seems to just be a restating of results and can be removed or shortened to make space for other discussion points, potentially around the final sentence on further study.

- **We shortened the sentence that re-iterated the results. However, we kept the remaining sentences as is, as they discuss and interpret data from the results section. For example, we mention that although our protocol called for 3 doses, only 2 doses might have been sufficient to achieve desired immunity.**

Line 186: It would be worth mentioning that in natural infection, Ureaplasma have immune evasion mechanisms, e.g. MBA variation. The difference(s) between naturally-acquired and vaccine-mediated immunity may be vast.

- **This has been added as a limitation (Lines 238-241)**

Can the authors comment on the relevance and impact of the infectious challenge used, compared to clinically identified infectious burden?

- **A direct comparison cannot reliably be made, given that in humans, bacterial burden is determined via sputum or bronchioalveolar lavage, where here, whole lung load per gram was quantified.**

Can the authors comment on what strategies they might take to achieve protection in 100% of infections, e.g. different adjuvant use, specific antigen use, etc. Or alternatively, provide some discussion about why 30% of animals were not protected?

- **Added (Lines 202-206)**

Figure 3: I'm not convinced that 'sham vaccinated' is the correct term here, as opposed to 'adjuvant only', which I think was the term used earlier in the text. As there was no inclusion of e.g. Ureaplasma growth media, to measure the effects of a sham immunisation, e.g. immunogenicity of the media.

- **“Sham” was replaced with adjuvant only.**

Additionally, can the authors provide evidence, e.g. statistical analysis, that control and immunised antibody measurements are significantly increased above pre-vaccination levels.

- **We favor not comparing different timepoints, and only focus on comparing control vs immunized mice. The reason is that we have run the assays for each time point (for both control and immunized) on different days. Also, the assay only generates relative antibody levels as opposed to absolute values. This makes comparisons between timepoints difficult and potentially unreliable.**

Please do a manuscript-wide proof-read to make sure abbreviations for UU and UP are consistent. E.g. in Figure 3, the difference between U. urealyticum and Ureaplasma parvum and Figure 4 where both are in full.

- **Done.**

Figure 4: can the authors please add a small amount more detail to the legend, e.g. what timepoint post infection tissue was harvested.

- **Added to the figure as recommended.**

For graph-based figures, please include the program (version) that graphs and statistics were produced in.

- **Added to each graph legend.**

Somewhere in the methods or figure legends, the authors need to indicate what graphs represent in terms of mean vs median, and SD vs SEM. It would also be beneficial to include error bars on graphs. It is unclear why a bar chart has been used in Figure 3, but not 4. However, this is a good use of displaying individual data points. Potentially a box and whisker plot would solve this issue, as it can be used to display (i) individual data points overlaid on summary data (the box), (ii) group mean, and (iii) error bars.

- **Figure legends were updated to include mean and SD. We have modified figure 3 (now figure 2) to show individual data points with error bars. We attempted a box and whisker plot with overlying data points but data was difficult to visualize clearly with that graph type.**

Reviewer #2 (Comments for the Author):

The major concern about this report is the study design. Only *U. urealyticum* was used for immunization while *U. parvum*, which is a more common colonizer in humans was left, making the whole study look like to be accomplished by half. The experimental grouping strategy looked confusing. It would also be interesting to test more strains within each species for infection to verify the specificity range of the immunity.

- **While it is true that *U. parvum* is a common human colonizer, *U. urealyticum* has been shown to be of equal if not greater prevalence in hyperammonemia cases (<https://doi.org/10.1093/ofid/ofad263>, <https://doi.org/10.1093/cid/ciaa1570>, <https://doi.org/10.1126/scitranslmed.aaa8419>). Because of this, the fact that there is no commercially available mouse Elisa kit for *Ureaplasma parvum*, and for ethical considerations about the large amount of additional animals that would need to be sacrificed for a *U. parvum* arm or testing on additional isolates, we focused on one clinically relevant *U. urealyticum* isolate. The lack of a *U. parvum* arm has been added as a limitation (Lines 244-247).**

Other comments:

1. Line 78: "12,500 rpm". Please provide the corresponding number of g forces.

- **This was an error. 12,500 was the RCF, not the RPM. This has been corrected.**

2. The description of the mouse vaccination and infection protocol is not clear and confusing. Were *U. urealyticum* and *U. parvum* infection protocols carried out the same time? If yes, why did they have different controls? It seems all 37 animals could be

better organized for grouping. Figure 1 and Figure 2 may be combined to show the whole experimental procedures. The time points when blood was drawn can be labeled in the combined figure to help the readers better understand Figure 3. In Figure 1, what's the meaning of "10x"? "1:1" can be deleted and instead, the exact numbers of male and female animals should be shown in each group.

- ***U. urealyticum* and *U. parvum* challenges were carried out at different time points, thus the different set of controls. This information was added to the methods section for clarification. Figures 1 and 2 were combined into one figure as recommended. Number of mice and female to male ratio were removed from the figure and clarified in the legends. Red arrows were added to the figure to indicate the time points when blood was drawn.**

3. For statistical analysis, please confirm if a normality test was performed prior to each designated statistical test.

- **Based on suggestions by both reviewers, all data was analyzed for normal distribution via the Shapiro-Wilk test, and for equal variance via Levene's test. For Figure 3 (relative plasma antibody levels), we found that while the data was normally distributed, the variances were not equal. Therefore, data was re-analyzed using Brown-Forsythe and Welch ANOVA tests with Dunnett's T3 multiple comparisons test, with individual comparisons computed for each pair. The figure, figure legend, and manuscript text has been updated accordingly. For Supplementary Figure 1, we found that the data was not normally distributed, and thus the Mann-Whitney test was used in place of the unpaired t-test. While the figure is unchanged, p-values have been updated. Notably, the updated statistical tests resulted in no changes in data interpretation throughout the manuscript.**

4. For all differences in *U. urealyticum* antibody levels, bacterial loads, blood anemia levels, are there any differences between male and female?

- **No significant differences between male and females were found.**

5. In Figure 3, the *U. urealyticum* antibody levels before infection in the two protocols are different. It looks like the background level (pre-vaccination) in panel B is about 2x of that in panel A. Are there any explanations for this phenomenon? Please also indicate the statistical significance test result between other group combinations, such as different doses groups, and immunosuppressed vs different doses. 6. What's the possible reason for the fact that bacterial loads of *U. parvum* were higher than *U. urealyticum* in control mice?

- **Part of the limitation of the assay is that it generates relative antibody levels as opposed to absolute numbers. We also have run assays for each time point on different days. This makes comparisons between timepoints difficult and potentially unreliable. Hence, we chose to focus only on comparisons within the timepoints between control and immunized, as those were completed on the same day in the same run.**

Re: Spectrum01763-24R1 (Prior immunity to *Ureaplasma urealyticum* protects against respiratory infection in immunosuppressed mice)

Dear Dr. Robin Patel:

Your manuscript has been accepted, and I am forwarding it to the ASM production staff for publication. Your paper will first be checked to make sure all elements meet the technical requirements. ASM staff will contact you if anything needs to be revised before copyediting and production can begin. Otherwise, you will be notified when your proofs are ready to be viewed.

Sincerely,
Artem Rogovsky
Editor
Microbiology Spectrum